# Copulation Duration and Sperm Precedence with Reference to Larval Diapause Induction in *Monochamus alternatus* Hope (Coleoptera: Cerambycidae)

**DOI:** 10.3390/insects15040255

**Published:** 2024-04-08

**Authors:** Katsumi Togashi, Hiroyuki Sugimoto

**Affiliations:** 1Laboratory of Forest Zoology, Graduate School of Agricultural and Life Sciences, University of Tokyo, Yayoi, Bunkyo-ku, Tokyo 113-8657, Japan; 2Forestry Engineering Department, Yamaguchi Agriculture and Forestry General Technology Center, Yamaguchi 753-0001, Japan; sugimoto.hiroyuki@pref.yamaguchi.lg.jp

**Keywords:** biological invasion, copulation duration, depletion of sperm, facultative diapause, *Monochamus alternatus*, obligate diapause, *P*_2_ value, pine sawyer, pinewood nematode, sperm precedence

## Abstract

**Simple Summary:**

Adults of the long-horned beetle *Monochamus alternatus* transmit a small worm, called the pinewood nematode, which causes pine wilt disease. A beetle subspecies in Taiwan (abbreviated ‘T’) has two generations a year (bivoltinism) due to facultative diapause, whereas another subspecies in Japan (abbreviated ‘J’) has a one- or two-year life cycle due to obligate diapause. T, with two infection periods a year, will cause more severe disease epidemics if it is introduced into Japan. Inter-subspecies hybridization is expected to inhibit the expression of bivoltinism because many hybrids induce diapause. To predict the effects of introducing T into Japan, the present study investigated copulation duration and the effects of the mating order of males on egg fertilization. The results indicated that a single copulation for more than 65 s supplied sufficient sperm to fertilize a lifetime production of eggs. When T females mated with a T male, the incidence of larval diapause for offspring was 0.15 and increased to 0.31 after females remated with a J male. Consequently, the proportion of second-male sperm used by T females was estimated to be 0.20. The effects of introducing T populations into Japan on the severity of disease epidemics were also discussed.

**Abstract:**

Adults of the pine sawyer *Monochamus alternatus* are the primary vector of *Bursaphelenchus xylophilus*, the causative agent of pine wilt disease. A sawyer subspecies in Taiwan (abbreviated ‘T’) has two generations a year (bivoltinism) due to facultative diapause, whereas another subspecies in Japan (abbreviated ‘J’) has a one- or two-year life cycle due to obligate diapause. T, with two infection periods a year, will cause more severe disease epidemics than J if it is introduced into Japan. Inter-subspecies hybridization may inhibit the expression of bivoltinism because many F1 hybrids induce diapause. To predict the effects of introducing T into Japan, the present study investigated copulation duration and late-male sperm precedence to fertilize eggs. The results indicated that a single copulation for more than 65 s supplied sufficient sperm to fertilize a lifetime production of eggs. The incidence of larval diapause was 0.15 for the offspring of T females that mated with a T male and increased to 0.292–0.333 after remating with a J male, while the incidence of larval diapause was 0.900–1.000 for hybrids from T females mated with a J male. Consequently, the estimated proportion of second-male sperm used by T females was 0.185–0.217. The effects of introducing T populations into Japan on the severity of disease epidemics were also discussed.

## 1. Introduction

Non-native species are being increasingly introduced into novel areas by the ever-growing international transport of people, resources, and products; this introduction of non-native species and the removal of barriers in habitats by human-mediated disturbances have resulted in hybridization between different taxa (e.g., [1,2,3,4,5]). Hybridization has been suggested to increase the invasiveness of hybridized lines (e.g., [1,2,3,4,5]) because hybrids may have enhanced fitness [6]. A novel gene combination (recombination), by which natural selection acts to generate a phenotype, may be better suited to colonizing novel environments [7]. It may lead to adaptive trait introgression [6]. Invasiveness may increase as a result of intraspecific hybridization among distantly related populations from different locations [1,2,8]. Invasiveness is defined as a measure of the ability of a species to enlarge its population size and spatial distribution with a lack of human assistance following its introduction [4]. 

In insects, sperm or spermatophores may be transferred from the male to the female (insemination) via the aedeagus during copulation, which involves the linking of male and female genitalia to form a firm connection between the two insects [9,10]. The amount of sperm supplied to females may vary depending on the duration of copulation (e.g., [11,12,13,14,15]). Sperm are stored in the spermatheca and used to fertilize the ova during a considerable period of time after copulation [9]. Therefore, in insect species whose females copulate with multiple males in a single bout of reproduction (multiple mating) [16], the production of intraspecific hybrids between distantly related populations varies depending on male–male competition for females, female receptivity, sperm competition, and cryptic female choice [10,16,17,18,19,20]. Sperm competition was originally defined as “the competition between sperm from two or more males for the fertilization of a given set of ova” [21] and was achieved by the aedeagus being inserted into female genitalia to flush or scrape the sperm of a rival from the spermatheca or push them to a distant area from the exit of the storage organ (sperm displacement) (e.g., [20,22,23,24]). The typical outcome of sperm competition is the last-male fertilization advantage (so-called last-male sperm precedence) (e.g., [10,20,22,23,24,25,26]).

On the other hand, females bias the relative paternity share among males during and after mating by their manipulation of sperm, such as expelling sperm from the tract, which is called the cryptic female choice [24,27,28]. Thus, last-male sperm precedence is a result of sperm competition and cryptic female choice and is expressed as the percentage of eggs fertilized by the last male (*P*_2_ value) [10,14,26,29]. The *P*_2_ value may be affected by a number of environmental conditions, such as the size distribution of males that attempt to copulate [29].

The Japanese pine sawyer *Monochamus alternatus* Hope is one of the primary vectors of the pinewood nematode *Bursaphelenchus xylophilus* (Steiner et Buhler) Nickle, the causative agent of pine wilt disease, which has been devastating pine forests in East Asia and West Europe [30]. This insect is widely distributed in East Asia and two subspecies have been identified: *M. a. alternatus* in China and Taiwan and *M. a. endai* Makihara in Japan and South Korea [31]. However, Lee et al. (2022) reported that it is impossible to separate *M. alternatus* into two subspecies using many specimens from China and South Korea and one or two specimens from Japan and Taiwan [32]. *M. alternatus* adults carrying *B. xylophilus* in the tracheal system emerge from dead host trees and feed on the bark of twigs from healthy host trees (e.g., [30]). After entering the host trees of susceptible species through insect feeding wounds, pathogenic nematodes kill them and then reproduce. Adults of *M. a. endai* are nocturnal: mating and oviposition occur at night and feeding occurs in the daytime [33,34]. Male adults mount the back of a female and copulate repeatedly while alternating between mount and half-mount (repetitive copulation) [34]. The duration of copulation was found to markedly vary between 3 to 100 s [34]; however, the relationship between the duration of copulation and the amount of sperm supplied to the partner currently remains unclear. *M. alternatus* females typically mate with different males (multiple mating) under outdoor conditions [34]. Reproductively mature females construct a slit or pit with their mandibles in the bark of recently dead trees, many of which are infected with *B. xylophilus*, insert the ovipositor through the wound, and generally lay none or one egg under the bark [35]. Larvae feed on the inner bark (phloem) and most make a pupal chamber in xylem [36,37]. They pupate and eclose into an adult in the chamber. Adults with *B. xylophilus* emerge from dead trees after screlotization of the cuticle. 

Two subspecies of *M. alternatus* exhibit different types of diapause: facultative diapause for *M. a. alternatus* and obligate diapause for *M. a. endai* [38,39]. Therefore, *M. a. endai* has a one- or two-year life cycle, whereas *M. a. alternatus* has two generations a year (bivoltinism), even when reared outdoors in Ibaraki Prefecture near Tokyo [39,40]. The bivoltine life cycle of vectors has two infection periods a year, leading to more severe epidemics of pine wilt disease than a univoltine life cycle. Although the induction of larval diapause in *M. a. alternatus* is primarily controlled by the photoperiod and temperature [38], it is suppressed by a reduction in the amount of food available and increased larval density [41,42]. The findings of a reciprocal cross-experiment indicated the dominance of diapause induction over nondiapause induction, the production of viable inter-subspecies hybrids, and the persistence of hybrid-derived populations [43]. Therefore, *M. a. endai* may prevent *M. a. alternatus* from expressing bivoltinism in Japan through inter-subspecies hybridization; *M. a. alternatus* is a potential threat to pine forests with regards to disease epidemics. Predictions based on a more detailed understanding of the copulation and sperm precedence of *M. alternatus* are needed.

Based on these findings, the present study investigated the following items: (i) the relationships between the duration of copulation, the fertilization of eggs laid by a female, and the depletion of stored sperm in *M. a. endai*; and (ii) the extent of last-male sperm precedence (*P*_2_ values) in *M. alternatus*, using the ecological trait of diapause or nondiapause induction. The effects of introducing *M. a. alternatus* into Japan on the severity of pine wilt disease epidemics were also discussed.

## 2. Materials and Methods

### 2.1. Insects

A laboratory population of *M. a. endai* was established from larvae in dead trees of *Pinus densiflora* Sieb. et Zucc. and *Pinus thunbergii* Parl. that had been harvested in Shika Town, Ishikawa Prefecture, Japan in 1990 (Appendix A). Eighteen female and 13 male newly emerged adults from generation 7 were used in Experiment 1. They were reared singly in transparent plastic cases (8.5 cm × 17.5 cm × 4.0 cm) after recording the emergence date, sex, body mass, and elytron length. Adults were identified by a combination of letters representing sex, F and M, and numbers. 

*M. a. alternatus* and *M. a. endai* adults were used in Experiments 2 and 3. A Taiwanese population of *M. a. alternatus* was established from four female and two male adults and six small feeding larvae that Dr. W. Toki had collected in Yangmingshan, Taipei City, Taiwan on 16 June 2010. This was performed with the permission of the Minister of Agriculture, Forestry and Fisheries, Japan (21Y1218). The laboratory population had been maintained over 9 generations (Appendix A). Nondiapause insects were used as parents in each generation. Adults that had forgone diapause in generations 8 and 9 were used in Experiments 2 and 3, respectively. The incidence of larval diapause was 0.000 and 0.030 in generations 8 and 9, respectively. Two other Japanese populations of *M. a. endai* collected in Abu Town, Yamaguchi Prefecture were used in Experiments 2 and 3. Dead trees of *P. thunbergii* damaged by *M. a. endai* larvae were felled in January 2014. Four female and 12 male adults that emerged from them were used in Experiment 2. Other dead trees of *P. thunbergii* were felled in December 2013. Offspring of the adults that emerged from them in 2014 were reared on *P. densiflora* bolts (Appendix A). Ten female and 30 male adults emerging from bolts were used in Experiment 3. All adults used in Experiments 2 and 3 were reared singly in transparent plastic cases (8.5 cm × 17.5 cm × 4.0 cm) with excised twigs of *P. densiflora* and *P. thunbergii*.

### 2.2. Pine Bolt, Branch and Twig Sections

Four healthy, and two healthy, trees of *P. densiflora* were harvested on 27 August and 21 October 2014, respectively, for Experiment 2. They were 9–14 years old and 4.81–8.05 cm in diameter at a height of 1.2 m. After being placed at room temperature for 4 or 5 days, stems were cut into 15-cm-long bolts. The cut ends of the bolts were coated with paraffin wax (melting temperature 56–58 °C) and then held at 10 °C until used. Three other healthy trees of *P. densiflora* were felled on 3 February 2015 for Experiment 3. They were 14 years old and 7.35–8.40 cm in diameter at a height of 1.2 m. Stems and branches were cut into 15-cm-long bolts and the cut ends of the bolts were coated with paraffin wax on 7 February 2015. Bolts were held at 25 °C until used.

No significant differences were noted in the mean or variance of bolt lengths, diameters, or bark surface areas between the five bolt groups used to inoculate the larvae that different subspecies of females produced after experiencing different types of mating in Experiment 2 (Appendix A). This was also the case for pine bolts used in Experiment 3 (Appendix A). The bark surface areas of bolts were more varied in Experiment 3 than in Experiment 2 to investigate whether the increasing effect of limited food availability on the induction of nondiapause in *M. a. alternatus* affected the estimation of the *P*_2_ value (the range of the bark surface area of bolts, 147.0–421.2 cm^2^ in Experiment 2, 73.0–461.8 cm^2^ in Experiment 3). 

Branch sections of *P. densiflora* were used as oviposition substrates. The mean lengths and diameters of the branch sections used were 8.9 cm (range 5.6–13.9 cm, n = 231) and 1.8 cm (range 1.4–2.7 cm), respectively, in Experiment 2 and 9.2 cm (range 6.2–13.1 cm, n = 47) and 1.5 cm (range 1.2–2.5 cm), respectively, in Experiment 3. To inhibit the oviposition behavior, thin 10- to 15-cm-long twig sections were provided for 24 h when *M. a. alternatus* females were allowed to remate with an *M. a. endai* male in Experiments 2 and 3 (mean diameter ± SE = 4.1 ± 0.3 mm, range 2–7 mm, n = 19). 

### 2.3. Larval Inoculation

A bark disc of ca. 8 mm in diameter was removed from a pine bolt using a carving knife and a depression was constructed on the surface of xylem or phloem. A first-instar larva, within 24 h of hatching, was placed in the depression and the bark disc was replaced in the original position. The disc was fastened with adhesive cloth tape.

### 2.4. Identification of Diapause and Larval Instars

When newly hatched larvae of *M. a. alternatus* were reared on *P. densiflora* and *P. thunbergii* bolts at 25 °C and a photoperiod of 16 h light and 8 h dark (LD 16:8 h), some emerged as adults between 70 and 139 days after the larval inoculation [41]. They were judged to be nondiapause insects, in other words, to have induced nondiapause. In contrast, other larvae remained in the bolts. Dissection of the bolts revealed two different types of living larvae and dead larvae *ca.* 150 days after the larval inoculation [41]. Yellowish white to yellow larvae with no fecal material in the intestines were considered to be in diapause because a long period of chilling allowed them to resume development [41,42,43]. When larvae had fecal material in the intestines, they were considered to be in the pre-diapause state [41,42,43]. This approach to distinguish between diapause and prediapause larvae was also used for *M. a. endai* [44]. When larvae were taken out of pine bolts, the larval instar was identified by the width of the head capsule: 1st instars = 0.585–1.170 mm wide, 2nd = 1.260–1.935 mm, 3rd = 1.980–3.150 mm, and 4th = 3.150–4.365 mm [45,46,47]. Dead pupae and adults found in bolts were judged to have induced nondiapause [39,41,42].

### 2.5. Relationship between the Duration of Copulation and Egg Fertilization (Experiment 1)

Nine virgin females that were 14 days old after emergence were placed individually into empty plastic cases (8.5 cm × 17.5 cm × 4.0 cm) and a male was then put on the back of the female at 25 °C under illumination (Table 1). Eight of the nine males that were paired individually with a female were virgins (mean age ± SE = 16.9 ± 1.2 days, range 14–25 days) and the other was a once-mated male aged 19 days; M12 was allowed to copulate with F13 4 days after copulating with F7. After being paired, close observations were performed to measure the duration of the first copulation using a stopwatch. The duration of copulation was defined as the time taken from the contact of the abdominal tips of both sexes to the male pulling the aedeagus out of the female abdomen. Immediately after the aedeagus was separated from the abdomen of the female, females were moved individually into plastic cases with a *P. densiflora* branch section (mean ± SE = 6.94 ± 0.02 cm for length, 2.15 ± 0.02 cm for diameter, n = 650) as the oviposition substrate and two or three thin, fresh, twig sections as food. They were reared at 25 °C and LD 12:12 h. Branch sections, which had been held at 25 °C for ca. 10 days to make them favorable for insect oviposition, were replaced daily with new ones and all eggs were removed using a small chisel and tweezers. Eggs were placed on wet tissue paper in Petri dishes, held at 25 °C and LD 12:12 h, and monitored daily for larval hatching. Eggs producing no larvae were dissected 10 days after oviposition. Eggs containing a dead larva, or a head capsule, were judged to be fertilized. When none of the 6 to 11 eggs laid initially were fertilized, females were paired again with the original partners for 3 h and eggs that were subsequently laid during the initial ca. 10 days were investigated for fertilization. On the other hand, when ten initially laid eggs produced larvae, females were reared until their death to investigate whether the depletion of stored sperm occurred. 

Eight other virgin females aged 14 days were allowed to copulate ten times individually with a male to measure the duration of copulation (Table 1). Thereafter, they were transferred singly into new plastic cases to investigate egg fertilization and the depletion of stored sperm in the same manner as that for females allowed to copulate only once. Of the eight males that were paired with each of the females, four were virgins (mean age ± SE = 18.5 ± 3.8 days, range 14–30 days) and four others were once-mated males (mean age ± SE = 19.0 ± 0.9 days, range 17–21 days); M3, M4, M9, and M22 were allowed to copulate with F4, F5, F12, and F17 on days 3, 3, 4, and 7 after their first mating, respectively. Furthermore, a 14-day-old virgin female was allowed to pair with a 15-day-old virgin male for 3 h and was thereafter investigated with regard to egg fertilization and the depletion of stored sperm (Table 1).

### 2.6. Second-Male Sperm Precedence (Experiments 2 and 3)

Experiments 2 and 3 were conducted to estimate the *P*_2_ value using a Taiwanese population of *M. a. alternatus* (sometimes termed ‘Taiwanese’ or ‘T’) and two Japanese populations of *M. a. endai* (sometimes termed ‘Japanese’ or ‘J’). In Experiment 2, four Taiwanese virgin females were paired individually with a Taiwanese virgin male in rearing cases with *P. densiflora* twig and branch sections (mating T [female] × T [male]) (Table 2). Branch sections were replaced with new ones every day. Eggs were taken out of branch sections every day and placed on wet tissue paper. Regarding each female, ten larvae that hatched early were inoculated singly on pine bolts during a period from 8 to 21 days after the start of pairing (Table 2). After completing the larval inoculation, the Taiwanese male was replaced with a Japanese virgin male, which was allowed to copulate with the Taiwanese female for 24 h in a rearing case in which there were thin twig sections of *P. densiflora,* but no branch sections as oviposition substrates. After the removal of the Japanese male, the Taiwanese female was provided with a pine branch section every day. Eggs were collected from branch sections 1–8 days after the removal of the Japanese male and larvae were prepared as stated above. Ten larvae that hatched early were inoculated individually on pine bolts during a period from 7 to 15 days after being paired with a Japanese male. Among eggs deposited during a period from 20 to 37 days after being paired with the Japanese male, 12 or 13 larvae that hatched early were also inoculated singly on pine bolts for three surviving females. In Experiment 3, ten Taiwanese females were used for this type of mating (Table 2). Three or four first-instar larvae were inoculated on pine bolts for each female after being paired with Taiwanese and Japanese males (Table 2). A few observations confirmed that some females copulated with Japanese males.

To assess the incidence of diapause for the estimation of *P*_2_ values, four other Taiwanese virgin females were paired individually with a Japanese male (mating T [female] × J [male]) in Experiment 2. To confirm obligate diapause in the Japanese beetle populations used in Experiments 2, four Japanese virgin females were paired individually with a Japanese male (mating J × J). Nine first-instar larvae that hatched early were inoculated singly on pine bolts for each female in T × J and J × J mating in the same manner as those in T × T mating in Experiment 2 (Table 2). In Experiment 3, ten females were used for each T × J and J × J mating (Table 2).

All larvae were reared at 25 °C, LD 16:8 h and 100% RH. Bolts with a larva were sometimes sprayed with water to keep the inner bark and xylem moist. Bolts were monitored daily to record the date of adult emergence. Bolts were dissected to assess the developmental stage of living and dead insects 156–173 days (mean ± SE = 167.0 ± 0.5 days) after the larval inoculation in Experiment 2 and 150–172 days (mean ± SE = 157.8 ± 0.7 days) in Experiment 3. When living larvae were obtained, records were kept of the width of their head capsule, body color, and the presence or absence of fecal material in the intestines to identify larval instar and diapause. 

Other experiments showed no effects of maternal age on the induction of offspring diapause in two local populations of *M. a. endai* [39] or in a strain of a Taiwanese population of *M. a. alternatus* that had been selected for the high incidence of diapause [48].

### 2.7. Estimation of P_2_ Values (Second-Male Sperm Precedence)

In Experiments 2 and 3, some Taiwanese females mated first with a Taiwanese male and then with a Japanese male; therefore, each of them carried sperm from the two males donated after remating with the second male. The *P*_2_ value (the second-male sperm precedence) is defined as the proportion of offspring sired by the second of two mating males (e.g., [10,25,26]). Therefore, after remating with Japanese males, Taiwanese females use sperm donated by a Taiwanese male at a probability of (1 − *P*_2_) and those by a Japanese male at a probability of *P*_2_ in order to fertilize an ovum. Using the diapause incidence (*d*_A_) of offspring produced by Taiwanese females that had only mated with a Taiwanese male and the diapause incidence (*d*_B_) of offspring produced by those that had only mated with a Japanese male, the average diapause incidence (*d*_D_) of offspring produced by Taiwanese females that had remated with a Japanese male after mating with a Taiwanese male was expressed as follows.
*d*_D_ = (1 − *P*_2_)*d*_A_ + *P*_2_*d*_B_.(1)

Therefore, the *P*_2_ value was estimated using the following equation. *P*_2_ = (*d*_D_ − *d*_A_)/(*d*_B_ − *d*_A_).(2)

To show the difference in the *P*_2_ value among females in Experiment 2, it was estimated for each of four Taiwanese females immediately after remating a Japanese male using Equation (2) with *d*_B_ = 1.00. When the estimate was negative, a value of 0.00 was assigned to the *P*_2_ value. The incidences of offspring diapause in Experiment 2 were calculated on the basis of 5–9 larvae that had induced diapause or nondiapause. However, the *P*_2_ value was not estimated for each Taiwanese female in Experiment 3 because the incidence of offspring diapause was calculated from the diapause response of 1–3 larvae.

### 2.8. Statistical Analysis

A one-way analysis of variance (ANOVA) was used to compare the length, diameter, and bark surface area of pine bolts. Their variance was compared using Bartlett’s test. Fisher’s exact test was used to compare the incidence of diapause. The Bonferroni method was used for pairwise comparisons of the incidence of diapause. The bimodality coefficient *BC* was applied to investigate whether the frequency distribution of copulation duration was unimodal or bimodal [49,50]. The coefficient is as follows:*BC* = (*γ*^2^ + 1)/[*κ* + 3(*n* − 1)^2^/{(*n* − 2) (*n* − 3)}], (3)
where *n* is the number of samples, and *γ* and *κ* are the skewness and excess kurtosis, respectively, of the frequency distribution. The *BC* value ranges between 0 and 1. The *BC* value is 5/9 (=0.555) for a uniform distribution and values ≥ 5/9 indicate a bimodal or multimodal frequency distribution.

## 3. Results

### 3.1. Relationship between Copulation Duration and Egg Fertilization (Experiment 1) 

Close observations of mating behavior indicated that each male mounted the back of a female and curled their abdomen downwards to contact the abdominal tip of the female. When allowed to insert the aedeagus into the female abdomen only once, M3, M9, M10, M12, and M15 touched their partner’s abdominal tip with theirs. When the abdominal tips were separated, the distal part of the semitransparent white aedeagus was found to remain in the female abdomen for a short or long time, indicating successful copulation. Males then pulled the aedeagus out of the abdomen of the female and retracted it into their abdomen. In contrast, M12, M21, M23, and M24 made contact two or three times with the abdominal tip of their partner for a short time, and the aedeagus was not observed between the two separating abdominal tips, indicating that it was not inserted (unsuccessful copulation). After unsuccessful copulation, these males successfully inserted the aedeagus. M21 successfully copulated with a walking female, whereas the others copulated with females that had stopped walking. 

A marked difference was observed in the duration of copulation among pairs that were allowed to copulate once (Table 3). A single copulation for less than 12.10 s did not enable females to produce fertilized eggs for virgin and once-mated males. Females with a short copulation were not sterile because they produced fertilized eggs after being paired with the original mate for 3 h (Table 3). In contrast, a single copulation for more than 63.96 s allowed females to produce fertilized eggs. Two females with a single long-term copulation deposited fertilized eggs during their lifetime, indicating no depletion of stored sperm (Table 3). A marked difference was also noted in the time interval between copulation and first oviposition (1–19 days), indicating that females received sperm from males through copulation even if they were not ready to oviposit (Table 3).

When males were allowed to insert the aedeagus ten times, two females were observed to refuse copulation by lashing the abdomen from side to side or putting the tip of the shrunken abdomen on the underside of the elytra. Six males did not successfully copulate 3–39 times by the time of the completion of ten successful copulations. After the aedeagus was pulled out of the female abdomen and retracted into the male abdomen, F17 excreted transparent fluid containing sperm from the abdominal tip and M22 had a small white mass containing sperm at the abdominal tip. Hereafter, the term copulation was exclusively used for successful copulation.

There were two peaks in the frequency distribution of copulation durations for eight females allowed to copulate ten times: 0 to 20 s and 60 to 70 s (*BC* value = 0.712 > 0.555) (Figure 1). Four of the eight females made ten short-term copulations of 1.70 to 36.93 s and produced no fertilized eggs (Table 4). They were not sterile because they laid fertilized eggs after being paired with their original partner for 3 h. In contrast, the four remaining females that copulated one or three times for more than 65 s produced fertilized eggs during their lifetime (Table 4). A female paired with a male for 3 h continued laying fertilized eggs during her lifetime (Table 4). A marked difference was noted in the time interval between copulation and first oviposition (1–10 days) (Table 4).

### 3.2. Second-Male Sperm Precedence (Experiments 2 and 3)

After mating with a Taiwanese male, four Taiwanese females produced 23 nondiapause and 5 diapause offspring in Experiment 2 (Table 5). Immediately after remating with a Japanese male, females produced 20 nondiapause and 10 diapause offspring. Females produced 29 nondiapause and 2 diapause offspring 20–26 days after remating with a Japanese male. Consequently, the incidence of offspring diapause increased from 0.148 to 0.333 immediately after Taiwanese females remated with a Japanese male and then decreased to 0.065 20–37 days after remating. In contrast, four other Taiwanese females produced diapause offspring alone when mating with a Japanese male (Table 5). Four Japanese females mating with a Japanese male also produced diapause offspring alone (Table 5). One fourth-instar and three third-instar larvae were in a prediapause state because of the presence of fecal material in the intestines at the time of bolt dissection (Table 5). 

A significant difference was observed in the incidence of offspring diapause among five groups of females that experienced different mating in Experiment 2 (Fisher’s exact test, *p* < 0.001) (Table 5). The incidence of offspring diapause was significantly lower in Taiwanese females that mated first with a Taiwanese male than in Taiwanese and Japanese females that mated with a Japanese male (Table 5). The *P*_2_ value was estimated to be 0.217 immediately after remating with a Japanese male, although there was a great difference in the *P*_2_ value among four females (mean ± SE = 0.255 ± 0.160, *P*_2_ = 0.00, 0.08, 0.22, 0.71). The incidence of offspring diapause slightly differed between Taiwanese females that mated with a Taiwanese male and those 20–37 days after remating with a Japanese male, suggesting *P*_2_ = 0.000. The overall *P*_2_ value was estimated to be 0.057, using the combined data of 49 nondiapause and 12 diapause offspring that females produced 1–8 and 20–37 days after remating (Figure 2).

In Experiment 3, after mating with a Taiwanese male, ten Taiwanese females produced 22 nondiapause and 4 diapause offspring (Table 6). Immediately after remating with a Japanese male, they produced 17 nondiapause and 7 diapause offspring. Consequently, the incidence of offspring diapause increased from 0.154 to 0.292 immediately after Taiwanese females remated with a Japanese male (Table 6). In contrast, ten other Taiwanese females produced 3 nondiapause and 27 diapause offspring when mating with a Japanese male (Table 6). Ten Japanese females that mated with a Japanese male produced diapause offspring alone (Table 6).

In Experiment 3, when the bark surface area of pine bolts was less than 41.0 cm^2^ (range of bolt diameters 1.7–1.9 cm), 78.6% of 14 inoculated larvae induced nondiapause: four of the four larvae produced by four Taiwanese females that mated with a Taiwanese male, four of the four larvae produced by four Taiwanese females that remated with a Japanese male after mating with a Taiwanese male, and three of the six larvae produced by six Taiwanese females that mated with a Japanese male. The incidence of diapause for inter-subspecies hybrids was significantly lower (3/6 = 0.500) when reared on thin pine bolts than on the other bolts (24/24 = 1.000) (Fisher’s exact test, *p* = 0.005).

A significant difference was observed in the incidence of offspring diapause among the four groups of females that experienced different mating in Experiment 3 (Fisher’s exact test, *p* < 0.001) (Table 6). The incidence of offspring diapause was significantly lower in Taiwanese females that mated first with a Taiwanese male than in Taiwanese and Japanese females that mated with a Japanese male (Table 6). The *P*_2_ value was estimated to be 0.185, which was similar to that in Experiment 2 (Table 6, Figure 2).

## 4. Discussion

The present study showed that there were two types of copulation in *M. alternatus*; long-term copulation for 64–109 s, during which sperm were transferred from males to females, and short-term copulation for less than 37 s, that was ineffective for sperm transfer. The results obtained also demonstrated that a single long-term copulation supplied a sufficient amount of sperm to fertilize all the eggs a female produced during her lifetime.

Males of the yellow-spotted longicorn beetle *Psacothea hilaris* showed two types of copulation: short-term (mean ± SD = 7.3 ± 3.3 s) and long-term (389.7 ± 49.3 s) insertion of the aedeagus [15]. In male mating behavior with previously mated (non-virgin) females, they repeated the short-time insertion of the aedeagus and then inserted it for a long time (long-term copulation) to transfer sperm [15]. During short-term copulation, males removed sperm from the spermatheca of the female using the aedeagus, which has a triangular process and pointed microbristles [23]. Consequently, the number of sperm stored in the spermatheca decreased to approximately 2% of that in the control females that had mated only once. We were unable to confirm whether long-term copulation was achieved at the end of a bout of repetitive copulations in *M. alternatus* because the present study suspended any repetitive copulation.

Multiple mating by female insects leads to competition among the sperm a female receives from different males [20]. A previous study reported marked variations in *P*_2_ values among species [10,25]. Sperm precedence of the second of two mating males is achieved in part by scraping or flushing the sperm of the first-mated male or setting the second-male sperm at the exit of the spermatheca using the aedeagus (e.g., [20,22,23,24]). In odonatans, the removal of sperm by scraping from the spermatheca is associated with higher *P*_2_ values [25]. Females of *M. alternatus* copulate with multiple males [34]. In the present study, we noted a small semitransparent-white jelly-like mass attached to the male abdominal tip. The jelly-like mass contained sperm. The results obtained indicated that the *P*_2_ value was ca. 0.2 immediately after the end of copulation by the second male and then slightly decreased with time. In contrast, the last-male sperm precedence is expected to be high in *P. hilaris* because males remove 98% of the sperm stored in the spermatheca with the aedeagus. The mechanisms responsible for low *P*_2_ values in *M. alternatus* have yet to be elucidated.

Fauziah et al. (1987) reported that after a male *M. a. endai* mounts the female and copulates, he maintains contact with his partner in a half-mount posture for an average of 208 min, during which he repeats 2–9 copulations and the mounted female lays eggs [34]. Each copulation lasts 3 to 100 s [34]. This mating behavior has also been reported for *Monochamus saltuarius* and *Monochamus scutellatus* [51,52]. *M. alternatus*, *M. saltuarius,* and *M. scutellatus* females also engage in multiple mating [34,51,52]. The benefits of postcopulatory mate-guarding are expected to increase at higher *P*_2_ values [53]. Postcopulatory mate-guarding is suggested to be an evolutionary stable strategy at extremely male-biased sex ratios, even when *P*_2_ values are low [53,54]. Males in some insects have been reported to intensify their mate-guarding behavior as the male-to-female ratio increases in order to prevent the access of other males to the ovipositing female [20,21,55,56,57,58]. *M. alternatus* adults congregate on dying and recently dead host trees to copulate and oviposit. Since females leave these trees following oviposition, the operational sex ratio will become male-biased. This has been reported in *M. scutellatus* populations on the most attractive trunk of cut *Pinus strobus* trees [59]. Therefore, postcopulatory mate-guarding is reasonable in *Monochamus* species, even if the *P*_2_ value (last-male fertilization advantage) is low.

The present study indicated that a single long-term copulation is sufficient to fertilize all the eggs a female *M. a. endai* produces during her lifetime (no depletion of stored sperm). Therefore, the females do not need to mate with multiple males. Furthermore, unmated females of a Taiwanese strain of *M. a. alternatus* exhibited significantly greater lifetime fecundity than females reared with a male during their lifetime [60], suggesting the harassment of females by males. However, multiple mating (copulation) in insects generally causes the lifetime fecundity of females to increase over single mating, reaching an intermediate optimum for the female mating rate, unless there are direct nutritional benefits [61]. The fitness of females that mated with a single male or multiple males may be separated into two types of benefits: non-genetic benefits (direct/first generation) and genetic benefits (indirect/second generation) [17]. Direct benefits result from the quality of the sperm of some males, which may increase female fecundity, longevity, or the mating rate, whereas indirect benefits are next-generation benefits that are connected to postcopulatory sexual selection mechanisms [17]. The former and latter benefits are evaluated using the numbers of offspring and grand-offspring, respectively, produced by a female [17]. However, the relationship between the number of males copulating with a female and the numbers of her offspring and grand-offspring has not yet been clarified for *M. alternatus*.

Low *P*_2_ values of approximately 0.2 immediately after copulation by the second males suggest that males gained enhanced fitness (mean number of offspring sired) from copulation with virgin females in *M. alternatus*. *M. a. endai* females have a very long preoviposition period because there were no well-developed ova in the ovaries at the time of emergence from host trees [62]. This study indicated that a female of *M. a. endai* made a single long-term copulation with a male 19 days before laying the initial two eggs. Reproductively mature females and males are attracted to and aggregate on declining, dying, or recently dead host trees in a pine forest on which they mate and oviposit [63,64,65,66,67]. Aggregation pheromones released by the males of *M. a. alternatus* and its congeneric species, as well as bark beetle pheromones acting as kairomones, may increase the likelihood of encountering virgin females [68,69,70,71,72,73].

Cryptic female choice may have an impact on the paternity share contingent upon sperm competition and, thus, affects the *P*_2_ value. Parker (2006) reported that cryptic female choice may occur at different levels, as follows: (i) ejaculate manipulation where females physically block the transfer of some sperm to sperm stores, eject them from the tract, or transfer them to a site where they are not used; (ii) sperm selection where females select against some sperm, either in the female tract or at the ovum surface; and (iii) differential reproduction investment, where females selectively bias the rate of egg-laying or the investment in offspring towards favored males [74]. It remains unclear whether the low *P*_2_ values in the present study are associated with cryptic female choice.

The bivoltinism of insect vectors will increase the severity of pine wilt disease epidemics over those with univolitinism because of the longer period of infection by the pathogenic nematode. Despite the dominance of diapause induction over nondiapause induction, a small proportion of F1 hybrids between *M. a. alternatus* females and *M. a. endai* males induced nondiapause, whereas the reverse cross-produced no F1 hybrids that induced nondiapause [43]. The present study showed that the limited availability of food to a larva caused inter-subspecies F1 hybrids to induce nondiapause. In other words, F1 hybrids induce nondiapause depending on environmental conditions.

When a population of *M. a. alternatus* adults emerge in late spring–early summer after arriving at a site in Japan, the number of offspring-generation adults that induce non-diapause and emerge in summer is closely related to the extent to which the severity of a pine wilt disease epidemic increases. If a population of *M. a. alternatus* arrives at a site where *M. a. endai* is at a relatively high population density, most females and males of *M. a. alternatus* will mate with different sexes of *M. a. endai*, resulting in the production of hybrids. High adult densities may increase larval densities in host trees killed by *B. xylophilus* [37]. Consequently, some hybrids may induce nondiapause [42], resulting in the higher mortality of pine trees. In contrast, if an *M. a. alternatus* population arrives at a site with a very low population density of *M. a. endai*, female adults emerging in late spring–early summer more often mate with males of the same subspecies and produce pure *M. a. alternatus* offspring that emerge as adults in the current summer. Furthermore, low *P*_2_ values indicate that already-mated *M. a. alternatus* females produce hybrids at low rates after remating with *M. a. endai* males, which results in an increased number of trees infected with the pathogenic nematode. If an *M. a. alternatus* population arrives at a site with an intermediate population density of *M. a. endai*, *M. a. alternatus* adults more frequently mate with *M. a. endai* and produce inter-subspecies hybrids. These hybrids induce diapause, possibly due to moderate larval densities in killed trees, suggesting no effect on disease epidemics. Therefore, the effects of introducing *M. a. alternatus* into Japan on pine wilt disease epidemics may vary, depending on the population density of *M. a. endai* at the arrival site.

## Figures and Tables

**Figure 1 insects-15-00255-f001:**
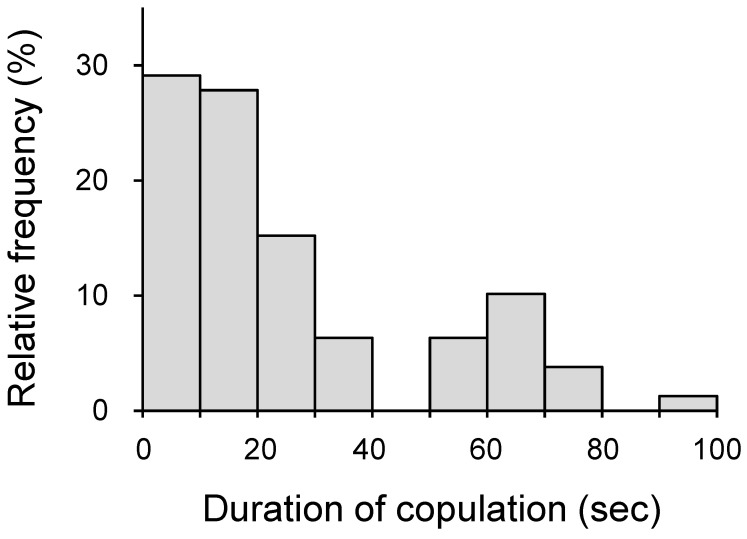
Frequency distribution of the copulation duration of *Monochamus alternatus endai*.

**Figure 2 insects-15-00255-f002:**
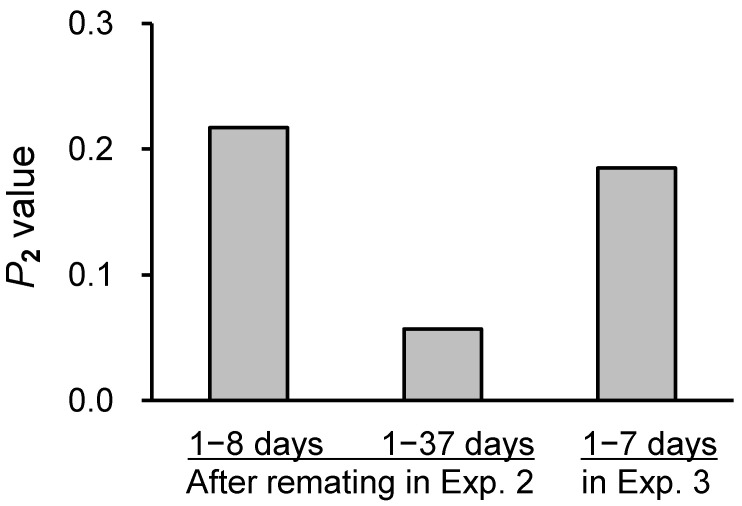
*P*_2_ values (second-male sperm precedence) shown by *Monochamus alternatus alternatus* females that remated with an *M. a. endai* male after mating with an *M. a. alternatus* male. The horizontal axis shows the time period during which females produced eggs after remating in Experiments 2 and 3.

**Table 1 insects-15-00255-t001:** *Monochamus alternatus endai* adults used in Experiment 1.

Number of Copulations Allowed	Number of Trials	Female		Male	
		Body Mass (mg) ^1^	Elytron Length (mm) ^1^	Body Mass (mg) ^1^	Elytron Length (mm) ^1^
1	9	394 ± 29	16.1 ± 0.5	348 ± 33	14.7 ± 0.5
10	8	359 ± 31	16.0 ± 0.5	347 ± 30	14.7 ± 0.3
>10 (3 h)	1	342	16.3	281	12.9

^1^ Mean ± SE.

**Table 2 insects-15-00255-t002:** Characteristics of *Monochamus alternatus alternatus* (T) and *M. a. endai* (J) adults used in Experiments 2 and 3 to assess the second-male sperm precedence (*P*_2_ value).

Type of Mating (Female × Male)	Characteristics	Experiment 2	Experiment 3
T females that mated with a T male (mating T × T)	Number of trials	4	10
	Age of females when paired (days)	37 or 38	17–38
	Age of males when paired (days)	20–43	13–47
	Number of larvae inoculated	40	31
	Time period from the start of pairing to egg collection (days) (Time period A)	1–14	–
	Time period from the start of pairing to larval inoculation (days) (Time period B)	8–21	8–30
T females that remated with a J male after first mating with a T male (mating T × J after mating T × T)	Number of trials	4	10
	Age of females when paired (days)	45–51	34–56
	Age of males when paired (days)	13–23	79–97
	Number of larvae inoculated immediately after being paired	40	31
	Time period from the end of pairing to egg collection (days) (Time period C)	1–8	1–7
	Time period from the end of pairing to larval inoculation (days) (Time period D)	7–15	7–13
	Number of larvae inoculated 29–43 days after being paired ^1^	37	–
	Time period C	20–37	–
	Time period D	29–43	–
T females that mated with a J male (mating T × J) ^2^	Number of trials	4	11
	Age of females when paired (days)	33–44	8–32
	Age of males when paired (days)	4–17	65–92
	Number of larvae inoculated	36	31
	Time period A	1–33	–
	Time period B	8–53	7–45
J females that mated with a J male (mating J × J) ^3^	Number of trials	4	10
	Age of females when paired (days)	22–26	61–75
	Age of males when paired (days)	15–22	61–73
	Number of larvae inoculated	36	31
	Time period A	2–20	–
	Time period B	8–27	7–19

^1^ Three surviving females oviposited. ^2^ The control experiment was implemented to estimate the *P*_2_ value. ^3^ The control experiment was implemented to confirm the obligate diapause in *M. a. endai*.

**Table 3 insects-15-00255-t003:** Relationships between the duration of the initial single copulation at the first pairing, the fertilization of eggs, and the depletion of stored sperm in virgin females of *Monochamus alternatus endai*. When being first paired with males, females were 14 days old after emerging from pine bolts. When depositing no fertilized eggs, each female was paired with her original partner for three hours to establish whether she was sterile. (Experiment 1).

Pair of Adults (Female Code × Male Code)	Duration of Copulation (s)	Female Age When the First Egg Was Laid (Days)	Time from Copulation to First Egg Laying (Days)	Period of Egg Collection after 1st Pairing (Days)	Number of Eggs Laid	% of Eggs Fertilized after 1st Pairing ^1^	% of Eggs Fertilized after 2nd Pairing ^2^
F1 × M3	12.10	15	1	10	9	0.0 (0/9)	100.0 (8/8)
F6 × M9	5.72	15	1	10	11	0.0 (0/11)	100.0 (27/27)
F7 × M12	5.36	15	1	10	11	0.0 (0/11)	66.7 (4/6)
F9 × M15	6.38	17	3	12	8	0.0 (0/8)	90.0 (18/20)
F13 × M12	1.99	23	9	18	8	0.0 (0/8)	94.7 (18/19)
F15 × M23	6.93	15	1	10	6	0.0 (0/6)	28.6 (2/7)
F18 × M21	5.38	25	11	21	8	0.0 (0/7)	100.0 (3/3)
F8 × M10	63.96	15	1	87 ^3^	106	97.1 (100/103)	― ^4^
F16 × M24	108.89	33	19	38 ^3^	12	100.0 (10/10)	― ^4^

^1^ Fertilization rate (number of larvae hatched/number of eggs laid). Larvae hatched include eggs with a developing embryo. Eggs laid exclude eggs killed by the female (biting) or the tweezers and chisel. ^2^ Fertilization rate (number of larvae hatched/number of eggs laid). Eggs were taken 3–11 days after females were paired with the original partners. ^3^ Investigated until the death of the female. ^4^ Females were not allowed to pair with their partners again.

**Table 4 insects-15-00255-t004:** Relationships between the duration of ten initial copulations at the first pairing, the fertilization of eggs, and the depletion of stored sperm in virgin females of *Monochamus alternatus endai*. When being first paired with males, females were 14 days old after emerging from pine bolts. When depositing no fertilized eggs, each female was paired with her original partner for three hours to establish whether she was sterile. (Experiment 1).

Pair of Adults (Female Code × Male Code)	Duration of First 10 Copulations (s) ^1^	Number of Long-Term Copula-tions ^2^	Female Age when the First Egg Was Laid (Days)	Time from Copulation to First Egg Laying (Days)	Period of Egg Collection after 1st Pairing (Days)	Number of Eggs Laid	% of Eggs Fertilized after 1st Pairing ^3^	% of Eggs Fertilized after 2nd Pairing ^4^
F2 × M4	18.54 ± 3.60 (5.66–35.35)	0	15	1	10	6	0.0 (0/6)	100.0 (9/9)
F4 × M3	13.55 ± 2.11 (2.91–26.85)	0	22	8	17	12	0.0 (0/11)	96.2 (25/26)
F5 × M4	7.28 ± 0.97 (1.70–11.10)	0	20	6	15	4	0.0 (0/4)	―
F19 × M17	19.22 ± 2.90 (7.32–36.93)	0	20	6	17	10	0.0 (0/10)	100.0 (4/4)
F10 × M20	37.76 ± 8.16 (5.88–73.97)	3	15	1	31 ^5^	28	88.0 (22/28)	― ^6^
F12 × M9	49.41 ± 9.56 (6.46–90.32)	3	15	1	65 ^5^	111	86.1 (93/108)	― ^6^
F14 × M22	29.48 ± 6.81 (3.70–73.80)	1	17	3	83 ^5^	110	95.0 (96/101)	― ^6^
F17 × M22	33.37 ± 8.10 (2.29–69.57)	1	24	10	72 ^5^	98	96.9 (95/98)	― ^6^
F3 × M5	Paired for 3 h	―	16	2	34 ^5^	49	100.0 (46/46)	― ^6^

^1^ Mean ± SE (range). ^2^ The duration of a long-term copulation is 65 s or longer. ^3^ Fertilization rate (number of larvae hatched/number of eggs laid). Larvae hatched include eggs with a developing embryo. Eggs laid exclude eggs killed by a female (biting) or the tweezers and chisel. ^4^ Fertilization rate (number of larvae hatched/number of eggs laid). Eggs were taken 2–11 days after females were paired with the original partners. ^5^ Investigated until the death of the female. ^6^ Females were not allowed to pair with their partners again.

**Table 5 insects-15-00255-t005:** Second-male sperm precedence (*P*_2_ value) and diapause incidence of offspring produced by *Monochamus alternatus alternatus* (T) and *M. a. endai* (J) females that mated with males of the same, and different, subspecies. (Experiment 2).

Female Characteristics	Number of Females Used	Number of Offspring						Incidence of Diapause (B/(A + B))	*P*_2_ Value Estimated
		Forgoing Diapause (A)	Undergoing Diapause (B)	In Pre-Diapause State	Dead in Larval Stage	Dead without Feeding ^1^	Total		
T females that mated with a T male	4	23	4	1	7	5	40	0.148 b ^2^	
T females 1–8 days after remating with a J male ^3^	4	20 (2) ^4^	10	0	7	3	40	0.333 b ^2^	0.217
T females 20–37 days after remating with a J male ^3^	3	29	2	1	0	5	37	0.065 b ^2^	
T females that mated with a J male	4	0	33	1	1	1	36	1.000 a ^2^	
J females that mated with a J male	4	0	28	1	5	2	36	1.000 a ^2^	

^1^ Failure of the larval inoculation. ^2^ Incidence of diapause followed by different letters differ from each other at the 5% significance level adjusted by the Bonferroni method after Fisher’s exact test. ^3^ Females first mated with a Taiwanese male before remating. ^4^ Number in parentheses shows the number of dead pupae found in bolts, which is included in the number of offspring that had forgone diapause.

**Table 6 insects-15-00255-t006:** Second-male sperm precedence (*P*_2_ value) and diapause incidence of offspring produced by *Monochamus alternatus alternatus* (T) and *M. a. endai* (J) females that mated with males of the same, and different, subspecies. (Experiment 3).

Female Characteristics	Number of Females Used	Number of Offspring					Incidence of Diapause (B/(A + B))	*P*_2_ Value Estimated
		Forgoing Diapause (A)	Undergoing Diapause (B)	Dead in Larval Stage	Dead without Feeding ^1^	Total		
T females that mated with a T male	10	22 (1) ^2^	4	5	0	31	0.154 b ^3^	
T females 1–7 days after remating with a J male ^4^	10	17 (1) ^2^	7	7	0	31	0.292 b ^3^	0.185
T females that mated with a J male	10	3 (1) ^2^	27	1	0	31	0.900 a ^3^	
J females that mated with a J male	10	0	28	2	1	31	1.000 a ^3^	

^1^ Failure of the larval inoculation. ^2^ Number in parentheses shows the number of dead pupae and adults found in bolts, which is included in the number of offspring that had forgone diapause. ^3^ Incidence of diapause followed by different letters differ from each other at the 5% significance level adjusted by the Bonferroni method after Fisher’s exact test. ^4^ Females first mated with a Taiwanese male before remating.

## Data Availability

All datasets in the current study are available from the corresponding author on reasonable request.

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
