# Peer review of "Copulation Duration and Sperm Precedence with Reference to Larval Diapause Induction in Monochamus alternatus Hope (Coleoptera: Cerambycidae)"

_insects, 2024, doi:10.3390/insects15040255_

Round 1

Reviewer 1 Report

Comments and Suggestions for Authors

 I encourage the authors to edit English language. 

Comments on the Quality of English Language

Author Response

Reviewer #1

The author demonstrate the influence of copulation duration and the number of copulations on egg fertilization, showing that a single copulation of more than 65s or multiple mating of female insects increase the rate of insemination. Meanwhile hybrids of T females and J populations scarcely induce nondiapause at intermediate densities of J populations. Study on T and J populations helps to predict the risk of the disease epidemics in Japan. However, the logic of this study is not rigorous, especially there are problems in experimental design and data analysis. I recommend a major revision, in which the following key points should be addressed.

Reply to reviewer #1: We greatly appreciate your careful comments. We rewrote the manuscript considering your comments.

  1. Line 23 mentioned that “hybrids scarcely induce nondiapause at intermediate densities of J populations”, and Line 396-397 mentioned that “Inter-subspecies hybrids expressed the induction of nondiapause due to insufficient amount of available food.” Are there a variety of factors contributing to non-diapause?

Reply: We deleted the sentence you pointed out (line 22-23, 36-37) because it was easy to make readers misunderstand what we want to express. The answer to your question and the reason for the deletion were as follows.

     The induction of diapause is controlled by photoperiod, temperature, the amount of food available and larval density [S1-S4]. This is added into the fourth paragraph of the Introduction section.

     This study (Line 396-397 in the previous manuscript) indicated that some inter-subspecies hybrids induced nondiapause only when the food available was extremely limited. The shortage of food is also caused by high larval densities in trees. Thus, we stated that the hybrids scarcely induce nondiapause at intermediate densities of J populations on line 23, because a small number of T beetles were considered to be introduced into Japan where J beetles were widely distributed. This consideration was made in the final paragraph of the Discussion section. But, it was difficult to state this consideration in Simple Summary and Abstract due to the limited number of words.

S.1. Enda, N.; Kitajima, H. Rearing of adults and larvae of the Taiwanese pine sawyer (Monochamus alternatus Hope, Coleoptera, Cerambycidae) on artificial diets. Trans. Annu. Meet. Jpn. For. Soc. 1990, 101, 503–504. (In Japanese)

S.2. Togashi, K. Effects of larval food shortage on diapause induction and adult traits in Taiwanese Monochamus alternatus alternatus. Entomol. Exp. Appl. 2014, 151, 34–42. [This article is cited in the manuscript.]

S.3. Togashi, K. Effects of crowding on larval diapause and adult body size in Monochamus alternatus alternatus (Coleoptera: Cerambycidae). Can. Entomol. 2017, 149, 159–173. [This article is cited in the manuscript.]

S.4. Togashi, K. Inverse density‑dependent diapause and its influence on population dynamics in Monochamus alternatus alternatus (Coleoptera: Cerambycidae). Appl. Entomol. Zool. 2021, 56:175–186.

  1. Line 185-187 mentioned the definition of diapause and pre-diapause state but how do you define Non-diapause. Although line 253-255 mentioned “body mass, width of head capsule, body color, the presence or absence of fecal material in the intestine were recorded to determine the diapause and larval instar” and the definition is not clear.

Reply: We added how to determine the nondiapause insects into subsection 2.4.

  1. In Table 2, the age of females used is different, and whether the age difference will affect the diapause rate of larvae, and then affect the experimental conclusion.

Reply: One of authors demonstrated no effect of maternal age on the induction of offspring diapause in a Taiwanese strain of M. a. alternatus which had been selected for a great incidence of diapause (Togashi, submitting). In the case of M. a. endai, the lack of maternal age effect on diapause induction has been already published in M. a. endai [S1, S2]. The information was added at the end of subsection 2.6.

S.1. Togashi K. Effects of photoperiod and chilling on diapause induction, intensity and termination in Monochamus alternatus endai (Coleoptera: Cerambycidae). Journal of Forest Research. 2019, 24(4), 243–249. [This article is cited in the manuscript.]

S.2. Togashi K. Influence of oviposition timing on offspring diapause and growth in Monochamus alternatus endai (Coleoptera: Cerambycidae). Journal of Forest Research. 2019, 24(5):313–319.

  1. In line 283-284, I don't think the algorithm of dD is correct, can you give relevant references?

Reply: Equation 1 expresses a weighted mean of a variable, when a data set is separated into two representing two components of a population, the means of which are dA and dB. So, this equation can be usually seen in any textbook of statistics. We revised how to obtain this equation considering your comment.

  1. Line 309, judging from the results, the number of eggs and hatching rate were detected after mating of 12.10s, 5.72s, 5.36s, 6.38s, 1.99s, 6.93s, 5.38s, 63.96s and 108.89s, respectively. The mating interval was relatively large, and the designed time gradient was unreasonable. Maybe the mating 50s also produces a lot of fertilized eggs. It is recommended to redesign the experiment and increase the experiment repetition.

Reply: The duration of copulation was not controlled artificially in Experiment 1. We did not interrupt the first copulation in the early half of Experiment 1. Each female was allowed to complete the first copulation and then separated from the partner. So, we add two words ‘the male’ to show who pulled the aedeagus in the sentence starting with “The duration of copulation was defined as the time taken…” in subsection 2.5 of the Materials and Methods section in the revised manuscript.

  1. Line 331-335 In Figure 1, error analysis and test repetition were not carried out, and the data were not convincing.

Reply: We added the results of statistical analysis following the comment by reviewer #3.

  1. Line358-359, according to the data of table 5, “from 0.179 to 0.333” should be changed to “from 0.148 to 0.333”

Reply: We greatly appreciate your indication. We corrected the numeral.

  1. Line 391-397, 78.6% of inoculated larvae induced non-diapause when food was insufficient. Is this specific experimental data available? Were each group treated with 78.6% inoculated larvae induced non-diapause?

Reply: The first sentence of this paragraph stated that 78.6% of 14 larvae inoculated on the thin pine bolts in Experiment 3 induced nondiapause. Phrases following ‘:’ indicate the numbers of nondiapause insects for each mating type.

     Experiment 3 included thinner bolts than Experiment 2 in order to determine whether the increasing effect of a small amount of food available on the induction of nondiapause in M. a. alternatus affected the estimation of the P2 value. We added the aim into the second paragraph of subsection 2.2 in the Materials and Methods section. In addition, we showed that there was no difference in the variation of bolt size between four bolt groups we used to rear larvae in Experiment 3. Please see Suppl. Table 1. 

  1. Line 400-402 mentioned “Pairwise comparison indicated that the incidence of diapause was significantly lower for Taiwanese females first mated with a Taiwanese male than for Taiwanese and Japanese females mated with a Japanese male (Table 6).” Table 6 shows that the incidence of diapause between Taiwanese females and Taiwanese males is 0.154ab, and that between Taiwanese females and Japanese males is 0.900ac. Is there a problem with the analysis of significant differences? There is the same problem in line 373-374.

Reply: We used a way to indicate two treatments between which there is a statistically significant difference. You can see this type of presentation in the sequential Bonferroni method. We explained this notation at the foot of Tables 5 and 6. However, we revised the notation of statistically significant difference considering your comment.

  1. Line 415-417, “the number of sperm stored in the spermatheca decreases to about 2% of that of control females which had mated only once.” I cannot find any data in the paper to support the result that the number of sperm in a spermatheca decreases to about 2%.

Reply: Yokoi (1990) tells “The number of sperm found in females’ spermathecae after the second stage copulation (long-term copulation) by the first male was 112.0×105±104.0×105 (per 1 ml, n = 10, mean ±SD). This number, however, decreased to 2.1×105±2.0×105 (per 1 ml, n = 11) just after first stage copulation (short-term copulation) by the second male.” Thus, the percentage of sperm removed from female spermathecae was 98% (= 100x[112.0x105-2.1x105]/112.0x105), although there were large differences between females (the first paragraph of the Results section, p. 384 in his article). This means that the number of sperm remaining in spermathecae decreases to 2% (= 100 – 98 %).

  1. Line 533, The reference section lacks literature from the past five years.

Reply: We tried to raise the number of recent references. Now, there are two references published after 2019 and 16 ones after 2014.

Reviewer 2 Report

Comments and Suggestions for Authors

This manuscript reported the observation of mating, the copulation duration, and the sperm precedence in Monochamus alternatus. Because this species carries nematodes causing the pine wilt disease, the reproductive ecology of M. alternatus is important as the authors mentioned. However, there are problems shown below in this manuscript. I cannot recommend publishing the manuscript without drastic revision and possibly additional experiments. In addition, the authors used the term “sperm precedence” as means of “second sperm precedence”. It may confuse readers because the term “first male sperm precedence” is also used in other articles.

L68 ")" was lacking.

L192 A male used in experiment 1 was once mated. In general, the mating history of males may affect the number of sperm transferred. The authors should analyze this sample separately in the results. In addition, the interval between his first and second mating should also be described.

L217 The mating interval of once-mated males may affect the number of sperm transferred generally. Thus, the mating interval should be mentioned in this paragraph.

L230 Did all insects mate with their partners for 24 hours? If authors observed their mating, the information should be described.

Table 5 The authors should show the P2-value for females 20-37 days after remating. I speculate that it is 0.0. Furthermore, why the authors did not include the results of 20-37 days after remating to calculate overall P2-values? 

Table 6 The authors should show the days after remating for females. The information helps readers to compare the results shown in Table 5.

Table 5-6 The histogram of P2-values should be shown.

L431 The mechanism of the low P2-values was discussed at L431. However, there is no evidence for this mechanism, and the explanation is insufficient. If the authors want to leave the sentence, additional explanations are required.

L441 The sentence contradicts the results. The present study showed that P2-values were low. In this case, first males may not be motivated to prevent other males.

L484 I disagree with the logic from L481 to L486. There is no evidence that M. alternatus shows potentially high P2-values, and the authors do not show any evidence that patterns of sperm precedence are similar among species in the same subfamily of insects.

L502 There is no evidence as basis for the discussion at L503. In the present study, P2 values when females mated with Japanese males for the first time and remated with Taiwanese males for the second time were uninvestigated. Since the order of mating may affect P2-values, experiments with reversed order are required. If cryptic female choice acts in this system as the authors mentioned, Taiwanese females may prefer Taiwanese males after mating.

Author Response

Comments and Suggestions for Authors (Reviewer #2)

This manuscript reported the observation of mating, the copulation duration, and the sperm precedence in Monochamus alternatus. Because this species carries nematodes causing the pine wilt disease, the reproductive ecology of M. alternatus is important as the authors mentioned. However, there are problems shown below in this manuscript. I cannot recommend publishing the manuscript without drastic revision and possibly additional experiments. In addition, the authors used the term “sperm precedence” as means of “second sperm precedence”. It may confuse readers because the term “first male sperm precedence” is also used in other articles.

Reply: We greatly appreciate your careful comments. We rewrote the manuscript considering your comments. However, we cannot carry out an additional experiment because of the elimination of Taiwanese populations in the laboratory following the law. We replaced “sperm precedence” with “second- or last-male sperm precedence” considering your comment.

L68 ")" was lacking.

Reply: We deleted the counterpart of the parentheses, “(“.

L192 A male used in experiment 1 was once mated. In general, the mating history of males may affect the number of sperm transferred. The authors should analyze this sample separately in the results. In addition, the interval between his first and second mating should also be described.

Reply: We rewrote the results considering your comment. We added the time interval between his first and second mating in this subsection (section 2.5).

L217 The mating interval of once-mated males may affect the number of sperm transferred generally. Thus, the mating interval should be mentioned in this paragraph.

Reply: We added the time interval between their first and second mating in this paragraph.

L230 Did all insects mate with their partners for 24 hours? If authors observed their mating, the information should be described.

Reply: I observed male behaviors several times in the photophase of photoperiod and confirmed that part of females copulated with Japanese males. We added it in this paragraph.

Table 5 The authors should show the P2-value for females 20-37 days after remating. I speculate that it is 0.0. Furthermore, why didn’t the authors include the results of 20-37 days after remating to calculate overall P2-values?

Reply: We added the speculation of P2 = 0.000 in the second paragraph of subsection 3.2. We also added the overall P2 value in the second paragraph of subsection 3.2.

Table 6 The authors should show the days after remating for females. The information helps readers to compare the results shown in Table 5.

Reply: We added the number of days after remating to make it possible to compare the results between Experiments 2 and 3.

Table 5-6 The histogram of P2-values should be shown.

Reply: We added Fig. 2 showing P2-values.

L431 The mechanism of the low P2-values was discussed at L431. However, there is no evidence for this mechanism, and the explanation is insufficient. If the authors want to leave the sentence, additional explanations are required.

Reply: We deleted the plausible mechanism of the low P2 values.

L441 The sentence contradicts the results. The present study showed that P2-values were low. In this case, first males may not be motivated to prevent other males.

Reply: You are right at a female-biased or even sex ratio. However, under a male-biased sex ratio, male postcopulatory guarding of partners is an evolutionary stable strategy (ESS) theoretically even when P2 values are low (Yamamura 1986). We added this reference and other empirical evidence. We added more explanation.

L484 I disagree with the logic from L481 to L486. There is no evidence that M. alternatus shows potentially high P2-values, and the authors do not show any evidence that patterns of sperm precedence are similar among species in the same subfamily of insects.

Reply: You are right. So, we deleted the late half of the paragraph and added another sentence.

L502 There is no evidence for the discussion at L503. In the present study, P2 values for females that mated with Japanese males for the first time and remated with Taiwanese males for the second time were uninvestigated. Since the order of mating may affect P2-values, experiments with reversed order are required. If cryptic female choice acts in this system as the authors mentioned, Taiwanese females may prefer Taiwanese males after mating.

Reply: You are right, although Taiwanese females were allowed to remate with a Japanese male after mating with a Taiwanese male in this study. We rewrote this paragraph considering your comment.

Reviewer 3 Report

Comments and Suggestions for Authors

Minor points

L118, L138

How many individuals you used for the experiment?

L334-L335

It is preferable that you should perform any statistical analysis if you say that there were two peaks in Fig 1 (This is closely related to your important result discussed at Discussion (L404-405).

Comment for Discussion

(preferable for revision when the two species of M. a. alternatus and M. a. endai are observed in spermatophore ejection and ingestion, or there is a possibility to occur them)

I am not familiar with the precise mating behaviours of these species. However, if they eject and ingest a spermatophore after mating, the related discussion (e.g. L452-L454) should be modified because females get direct nutritional benefit from multiple mating and multiple-mating is beneficial both for male and female.

References for further discussion

Obata S., Hidaka T. (1987) Ejection and ingestion of the spermatophore by the female ladybird beetle, Harmonia axyridis Pallas (Coleoptera: Coccinellidae).

The Canadian Entomologist 119: 603-604. DOI: https://doi.org/10.4039/Ent119603-6

Perry J.C., Row L. (2008) Ingested spermatophores accelerate reproduction and increase mating resistance but are not a source of sexual conflict. Animal Behavior 76: 993-1000. doi:10.1016/j.anbehav.2008.05.017

Thornhill R., Alcock J. (1983) The Evolution of Insect Mating Systems. Cambridge, Harvard University Press. https://doi.org/10.4159/harvard.9780674433960

Author Response

Comments and Suggestions for Authors (Reviewer #3)

Reply to reviewer #3: We greatly appreciate your invaluable comments. We rewrote the manuscript considering your comments.

Minor points

L118, L138

How many individuals you used for the experiment?

Reply: We added the numbers of adults used in experiment 1.

L334-L335

It is preferable that you should perform any statistical analysis if you say that there were two peaks in Fig 1 (This is closely related to your important result discussed at Discussion (L404-405).

Reply: We used a statistical method to show a bimodal frequency distribution.

Comment for Discussion

(preferable for revision when the two subspecies of M. a. alternatus and M. a. endai are observed in spermatophore ejection and ingestion, or there is a possibility to occur them).

I am not familiar with the precise mating behaviours of these subspecies. However, if they eject and ingest a spermatophore after mating, the related discussion (e.g. L452-L454) should be modified because females get direct nutritional benefit from multiple mating and multiple-mating is beneficial both for male and female.

Reply: We greatly appreciate your suggestion. However, to our knowledge, nobody has reported female ejection and ingestion of a spermatophore after mating. So, we did not add the topic of spermatophore consumption.

References for further discussion

Reply: I greatly appreciate your suggestion of invaluable references. However, the two former references were not cited because we did not touch the consumption of spermatophores. We have already cited the final reference in this manuscript.

Obata S., Hidaka T. (1987) Ejection and ingestion of the spermatophore by the female ladybird beetle, Harmonia axyridis Pallas (Coleoptera: Coccinellidae). The Canadian Entomologist 119: 603-604. DOI: https://doi.org/10.4039/Ent119603-6

Perry J.C., Row L. (2008) Ingested spermatophores accelerate reproduction and increase mating resistance but are not a source of sexual conflict. Animal Behavior 76: 993-1000. doi:10.1016/j.anbehav.2008.05.017

Thornhill R., Alcock J. (1983) The Evolution of Insect Mating Systems. Cambridge, Harvard University Press. https://doi.org/10.4159/harvard.9780674433960

Round 2

Reviewer 1 Report

Comments and Suggestions for Authors

The author has answered and modified the questions I raised, and the experimental design is appropriate to test the hypothesis. 

Comments on the Quality of English Language

The English improvement is required

Author Response

We are greatly appreciate your suggestions. We revised the manuscript considering the suggestions. Corrected phrases and sentences in the revised manuscript are shown in red.

  1. We added some sentences and made some corrections in subsection 2.4 of the Materials and Methods section and in Tables 5 and 6 of the Results section.

     Dead pupae and adults found in bolts were included in insects that had induced nondiapause in Experiments 2 and 3. This was added in subsection 2.4 of the Materials and Methods section and in Tables 5 and 6 of the Results section.

  1. Relies to your Comments and Suggestions

The author has answered and modified the questions I raised, and the experimental design is appropriate to test the hypothesis.

Reply: We greatly appreciate your comments and suggestions.

Comments on the Quality of English Language

The English improvement is required.

Reply: A company named “Medical English Service” improved English of the revised manuscript we submitted previously. For example, they deleted “s” from “incidences” in a sentence within footnotes of Tables 5 and 6, although I thought the correction was strange considering the grammar. You stated that minor editing of English language is required. Your statement means a small number of sentences we should improve. So, if you pointed out strange English expression, we would ask them to improve sentences and phrases.

Reviewer 2 Report

Comments and Suggestions for Authors

The manuscript was well-revised mostly. 

I understand the difficulty of additional experiments.

However, Fig. 2 should be revised again because the figure is not a histogram that is a standard style indicating P2-values in the study of sperm competition.

Histograms help us to distinguish monomodal and bimodal distributions.

For example, 7 and 10 females were used to estimate P2 values in experiments 2 and 3, respectively.

Therefore, I recommend the authors show the distribution of P2 values in the experiments as a histogram.

Minor point

L536 There is unnecessary "-" at L536.

Author Response

We are greatly appreciate your suggestions. We revised the manuscript considering the suggestions. Corrected phrases and sentences in the revised manuscript are shown in red.

  1. We added some sentences and made some corrections in subsection 2.4 of the Materials and Methods section and in Tables 5 and 6 of the Results section.

     Dead pupae and adults found in bolts were included in insects that had induced nondiapause in Experiments 2 and 3. This was added in subsection 2.4 of the Materials and Methods section and in Tables 5 and 6 of the Results section.

  1. Relies to your Comments and Suggestions

The manuscript was well-revised mostly.

Reply: We greatly appreciate your comments and suggestions.

I understand the difficulty of additional experiments.

Reply: We greatly appreciate your understanding.

However, Fig. 2 should be revised again because the figure is not a histogram that is a standard style indicating P2-values in the study of sperm competition.

Reply: We cannot show a frequency distribution of P2 values because we got only four P2 values. So, we added the mean and SE of the four P2 values into subsection 3.2 of the Results section.

Histograms help us to distinguish monomodal and bimodal distributions.

Reply: If the P2 values are estimated for many females, we will have a frequency distribution of P2 values. However, we could estimate only four P2 values. So, we did not depict a frequency distribution of P2 values.

For example, 7 and 10 females were used to estimate P2 values in experiments 2 and 3, respectively.

Reply: The T x J mating after mating with a T male is necessary to estimate P2 value. In Experiment 2, four females were used to obtain the incidence of offspring diapause after T x T mating and T x J mating after mating with a T male. In Experiment 2, the diapause response of 5–9 offspring was used to calculate the incidence of diapause for each of the four Taiwanese females. Thus, we could estimate P2 value for each Taiwanese female. In Experiment 3, the induction of offspring diapause for each female would be calculated on the basis of 1–3 offspring. The number of offspring was too small to estimate accurate incidence of diapause. We did not calculate P2 values for ten females in Experiment 3.

Therefore, I recommend the authors show the distribution of P2 values in the experiments as a histogram.

Reply: You are right. However, we cannot depict the frequency distribution of P2 value.

Minor point

L536 There is unnecessary "-" at L536.

Reply: We did not delete ‘-’ from a phrase ‘late spring-early summer’, because ‘-’ means ‘to’. If ‘-’ is deleted, a phrase of ‘late spring’ becomes an adjective of ‘early summer’.